# Looking for a Tailored Therapy for Heart Failure: Are We Capable of Treating the Patient Instead of the Disease?

**DOI:** 10.3390/jcm10194325

**Published:** 2021-09-23

**Authors:** Alessandro Fucili, Paolo Cimaglia, Paolo Severi, Francesco Giannini, Alberto Boccadoro, Marco Micillo, Claudio Rapezzi, Luigi Tavazzi, Roberto Ferrari

**Affiliations:** 1Cardiovascular Research Centre, Ferrara University, Via Aldo Moro 8, 44124 Ferrara, Italy; a.fucili@ospfe.it (A.F.); p.severi@hotmail.it (P.S.); alberto.boccadoro@gmail.com (A.B.); marcomicillomd2@outlook.it (M.M.); claudio.rapezzi@unibo.it (C.R.); 2Cardiovascular Department, GVM Care and Research, Maria Cecilia Hospital, Via Corriera 1, 48033 Cotignola, Italy; paolocimaglia88@gmail.com (P.C.); giannini_fra@yahoo.it (F.G.); ltavazzi@gvmnet.it (L.T.); 3Via Ercole I° D’Este 32, 44124 Ferrara, Italy

**Keywords:** heart failure, beta-blockers, ivabradine, ACEi, angiotensin-converting enzyme inhibitor, ARBs, angiotensin receptor blockers, MRA, mineral corticoid receptor antagonist

## Abstract

After almost a decade of stagnation in clinical research for HF treatment, five large randomized trials recently published have supported the use of four new classes of drugs, namely: angiotensin receptor/neprilysin inhibitor, sodium–glucose co-transporters 2 inhibitors, soluble guanylate cyclase modulators, and myosin activators. Each treatment has proved to be beneficial for both long-term outcomes and quality of life. Beside their clinical relevance, all these novel treatments have a different mechanism of action beyond the usual neuro-hormonal blockage. These different pathways, together with the unquestionable clinical evidence, advocate a re-thinking of HF treatment and of the appropriate drug to integrate with the existing standard therapy, according to different characteristics of HFrEF patients. This study aimed to offer a synthetic overview of the mechanisms of action of the new drugs and to propose a more personalized approach, considering patients’ characteristics and safety profiles. To this end, we have identified seven profiles for patients with chronic heart failure with reduced ejection fraction and two for pre-discharge patients.

## 1. Introduction

In the last six years, “*A New Era*” has emerged for medical treatment of heart failure with reduced ejection fraction (*HFrEF*). After the 1970s, when heart failure (*HF*) was empirically treated with diuretics, digitalis, and bed rest, a revolution started based on the understanding that, for the failing heart, long-lasting neuroendocrine activation is deleterious rather than beneficial [1]. As a consequence, evidence that several neuro-hormonal antagonists exert a striking prognostic effect emerged [2]. Then, a period of, at least, a decade of silence for both basic and clinical researches followed. Several randomized clinical trials (*RCTs*) exploring new treatments provided, at best, neutral results. In 2014, the scientific community started to be resigned and to believe that the limit of HF knowledge was reached.

Since 2014, five landmark randomized clinical trials (RCTs) have reported unexpected favorable results in HFrEF from four new classes of drugs acting beyond the classical neuro-hormonal blockage. These new drugs, namely, angiotensin receptor/neprilysin inhibitors (*ARNI*), sodium–glucose transporter 2 inhibitors (*SGLT2i*), sodium guanylate cyclase (*SGC*) modulators, and myosin activators, have shown to further increase survival and/or hospitalization for patients with HFrEF and are, or soon will be, available for HFrEF treatment. In anticipation of the new guidelines, several suggestions on how to integrate these treatments have been proposed [3,4,5,6,7]. The suggestions are based on the relative efficacy of each drug to achieve the pre-determined endpoints, with the aim to produce an integrated scheme related to the use of the new treatments in clinical practice.

As often is the case, the new RCTs are not comparable: they were conducted in different periods, with different background therapies, and have enrolled patients with different characteristics as shown in Table 1. Thus, it is difficult to extrapolate just from these trials the actual “*value to cure*” of each drug and to suggest a standardized framework good for all patients. The difficulty is amplified in the “*real world*” given the reluctance of practitioners to adhere to schematic guidelines’ recommendations for multiple reasons, including tolerability, which usually is related to low blood pressure, heart rate, poor kidney function, or hyperkalemia. This is particularly so in elderly patients with several comorbidities and multiple pharmacotherapies [2,3,4,5,6,7,8].

In this review, we followed a different approach. As the four new classes of drugs produced positive results, although with different degrees of evidence, we proposed to integrate them with the existing standard therapy according to the patients’ characteristics, which is what practitioners are requested to do, particularly in elderly HF patients, which are the majority. We have identified several HFrEF phenotypes common in the “*real world*” and identified the most appropriate drug to be used, based on the inclusion and exclusion criteria of the RCTs. Our suggestion, outlined graphically in Figure 1, Figure 2 and Figure 3, considered specific patient profiles to stimulate physicians to personalize their treatments according to the evidence and to pathophysiological and clinical reasoning. To achieve this, an understanding of the mode of action of the new classes of drugs is needed.

## 2. Angiotensin Receptor/Neprilysin Inhibitor (ARNI)

ARNI represents a change of paradigm in HFrEF treatment. This class of drugs combines RAAS inhibition through angiotensin II receptor blockage (*by valsartan*) to the inhibition (*by sacubitril*) of a new target, neprilysin, a ubiquitarian enzyme that degrades several peptides, including endogenous natriuretic peptides (*ANP*). As a result, the deleterious vasoconstriction and water retention induced by angiotensin II are counteracted, whilst the beneficial vasodilatation and diuretic action of natriuretic peptides are emphasized [9]. Thus, for the first time, the neuroendocrine response is considered as a whole: it is not only deleterious, but it can be beneficial. Instead of just antagonizing it, ARNI further recruits the beneficial vasodilatory and diuretics action of ANP and blocks the dangerous effect of angiotensin II. In so doing, ARNI reinstates a more physiological neuroendocrine balance resulting in amelioration of physiological diuresis, and reduction of peripheral resistance and blood pressure with improvement of symptoms and outcomes as shown by the PARADIGM-HF trial [10]. Sacubitril/valsartan caused a 20% reduction in cardiovascular (*CV*) death and a 21% reduction in hospitalizations for HF (*HHF*) compared to enalapril in symptomatic HFrEF patients with left ventricular ejection fraction (*LVEF*) ≤40% (*subsequently amended to* ≤35%) and high BNP/NT-proBNP. Clinical benefit was rapid as shown by the Kaplan–Mayer curves for primary outcome, which start to diverge already after the first month. This was, in part at least, attributable to reverse cardiac remodeling, with improvement in left ventricular function concomitant to early reduction in NT-proBNP levels [11]. The safety profile was favorable for sacubitril/valsartan, as enalapril was discontinued more often than sacubitril/valsartan. The most frequent adverse event was symptomatic hypotension, rarely requiring discontinuation of the study drug. Creatinine elevation above 2.5 mg/dL and serum potassium ≥ 6 mmol/L were significantly less frequent in ARNI’s arm compared to enalapril [10]. Following the design of PARADIGM-HF, which included a running phase before screening with patients required to take a stable dose of ACEi or angiotensin-II receptor blockers (ARBs) equivalent to at least 10 mg of enalapril daily, the guidelines of European Society of Cardiology (ESC) 2016 recommend the use of ARNI in symptomatic heart failure with reduced ejection fraction (HFrEF) patients despite 3 to 6 months of therapy with full dose of ACE inhibitors (ACEi), beta-blockers (BB) and mineral corticoid antagonist (MRA) [12]. Thus, this resembles the design of PARADIGM-HF.

The program continued with the PIONEER-HF trial [13]. Administration of sacubitril/valsartan in hospitalized patients for acute HF with EF ≤ 40% showed reduction in NT-proBNP concentration greater in the active arm compared to enalapril. The differences between groups were already evident after 1 week, even in patients with de novo acute HF untreated with ACEi or ARBs [13]. The TRANSITION trial confirmed the safety profile of sacubitril/valsartan administered to patients with acute decompensated HF. Some of these patients were newly diagnosed and were not receiving ACEi/ARBs [14]. A recent sub-analysis of patients with ICD enrolled in the PARADIGM-HF showed that treatment with sacubitril/valsartan reduced sudden cardiac death independently from the ICD status in the ARNI arm [15]. Other reassuring data come from two small studies: the PROVE-HF and EVALUATE-HF. Both studies showed a powerful reverse remodeling action of sacubitril/valsartan [11,16]. Of note, PARADIGM did not follow the usual add-on scheme for all the other HF trials but, for the first time, it compared a standard therapy with enalapril to sacubitril/valsartan and established the superiority of the latter treatment.

These considerations advocate for an early treatment initiation strategy for sacubitril/valsartan in chronic HFrEF patients, independently from whether receiving ACEi/ARBs or not. This is also the opinion of the “*2021 Update to the 2017 ACC Expert Consensus Decision Pathway for Optimization of Heart Failure Treatment*” that defined ARNI as the preferred renin–angiotensin antagonist in HFrEF [17]. Several experts share the same opinion [2,3,4,5,6,7].

## 3. Sodium–Glucose Co-Transporters 2 Inhibitors (SGLT2i)

SGLT2i are a new class of drugs originally created and approved as antidiabetic agents for the treatment of type 2 diabetes mellitus. SGLT2i promote urinary glucose excretion by inhibition of glucose and sodium reabsorption from the renal proximal tubules where SGLT2_,_ the glucose co-transport protein, is located [18]. To fulfil the US Food and Drug Administration (*FDA*) requirements for new antidiabetic drugs, since 2008 three major cardiovascular (CV) outcome trials were performed with SGLT2i to rule out CV harm, while improving glucose control. Unexpectedly, all these trials showed positive outcomes in terms of CV death, representing a real case of serendipity [19]. These observations set the stage for two further RCTs in patients with confirmed and well-treated, the DAPA-HF and the EMPEROR-Reduced trials [20,21].

The DAPA-HF trial showed a 26% reduction in the primary endpoint of CV mortality and HHF or urgent HF visits in patients with LVEF ≤ 40% and high BNP/NT-proBNP, regardless of the diabetic status. As a result, in 2020, the FDA approved the SGLT2i dapagliflozin for the treatment of patients with HFrEF [22]. One year later, the EMPEROR-Reduced trial confirmed the benefit of SGLT2i empagliflozin in HFrEF with a 25% reduction in the primary endpoint (*CV death and HHF*) in treated patients [21]. The primary endpoint was driven by HHF and the reduction of CV death was not statistically significant. This could be explained in terms of too-small sample size, 4744 vs. 3660 patients for DAPA-HF vs. EMPEROR-Reduced, respectively, and low statistical power of EMPEROR-Reduced for detecting CV mortality. A recent meta-analysis shows a statistically significant reduction of CV mortality, combining data from both the DAPA-HF and the EMPEROR-Reduced trials [23].

In both studies, benefits occurred early, after receiving the active drugs regardless of background evidence-based therapy but with no more than 20% of patients on ARNI. The data confirmed significantly better results for both SGLT2i in all the predefined groups with clear amelioration of kidney functions and outcome and excellent tolerability. A cross-sectional analysis demonstrated a 1.4 to 6.3 years gain of life expectancy with comprehensive (*ARNI, BB, MRA, and SGLT2i*) vs. conventional (*ACEi, BB, MRA*) therapy [24]. Contrary to ARNI, some evidence on safety and efficacy of SGLT2i was also available for the acute setting. The SOLOIST trial, demonstrated that treatment with sotagliflozin reduced the risk of CV deaths and HHF/urgent HF visits in patients with acute HF by 33% [25]. However, all enrolled patients were diabetic, there was no LVEF cut off for inclusion, with 79% of patients showing an LVEF < 50%, and the study was underpowered because it prematurely stopped. Thus, care should be used in extrapolating SOLOIST results to all SGLT2i, also considering that sotagliflozin is a less selective SGLT2i.

The mechanism of action of SGLT2i in HFrEF is still a mystery with several questions unraveled. What is clear is that the scientific community is missing several pieces of the HF puzzle. SGLT2i are efficacious independently from neuro-hormonal antagonism, leaving a great deal to learn and to explore. The liturgy of drug development is reversed in the case of SGLT2i: unquestionable evidence is now searching for a plausible explanation [19]. Hypotheses are not lacking. Several mechanisms have been proposed including antifibrotic, anti-inflammatory, antioxidative, and antiapoptotic properties, as well as improved hemodynamic and ventricular unloading due to reduction of blood pressure, arterial stiffness, and weight loss [26]. Other explanations relate to reduction of uric acid levels, and of epicardial adipose tissue with beneficial paracrine regulation of adipokines and reduced serum leptin levels [24]. Improved cardiac energy by increased glucagon utilization and shifting metabolism from fatty-acid oxidation to glucose has also been considered [27]. Another possibility is a direct cardiac effect through inhibition of sodium–hydrogen exchange which, in turn, may lead to reduction in cardiac remodeling [26]. However, the timeframe and the magnitude of clinical effects of SGLT2i exclude several, if not all, of the above-mentioned mechanisms of action. At present, the most logical and obvious target remains the kidney where, besides the inhibition of the co-transporter, SGLT2i reduce intraglomerular pressure due to a tubule–glomerular feedback resulting in loop diuretic sparing effect. SGLT2i induce an osmotic diuresis with glycosuria, albuminuria, and uricosuria, making these drugs not just a classical diuretic, which, by the way, in HF are not associated with CV benefits. SGLT2i cause a sort of “*pharmacological ultrafiltration*” with important reduction of volume overload and improvement of ventricular function. A profitable vicious circle between the heart and the kidney is therefore established, contrasting so-called and, often underestimated, cardiorenal syndrome. A confirmation of this hypothesis comes from the recent CREDENCE and DAPA-CKD trial, which showed a reduction of risk progression of kidney disease and of HF in diabetic and non-diabetic patients [28,29].

Thus, as for ARNI, there is ample evidence for an early introduction of SGLT2i in the treatment of HFrEF patients. As the mechanism of action of SGLT2i is independent from a reduction of neuroendocrine activation, these drugs, when possible, should be used in combination with standard neuro-hormonal antagonists or ARNI [2,3,4,5,6,7].

## 4. Soluble Guanylate Cyclase (SGC) Modulators

The nitric-oxide-guanylate cyclase (SGC)-CGMP is a complex signal transduction mechanism exerting a key role in the regulation of the CV system. Central to these pathways is *SGC*, an intracellular enzyme which is expressed in the different components of the heart, including cardiomyocytes, smooth muscle cells of blood vessels, and platelets [30]. SGC has a high affinity for nitric oxide (*NO*) which, in turn, is synthesized by three nitric oxide synthases (*NOS*) from L-arginine. In endothelial cells, NO is produced by eNOS. being a gas, NO rapidly reaches the smooth muscle cells and binds to SGC, its target enzyme. The binding activates SGC which, in turn, is responsible for the release of the second messenger cGMP. cGMP causes dilatation through three intracellular pathways, such as cGMP-dependent protein kinases, cGMP regulated ion channels, and phosphodiesterases. All the three transduction pathways exert different effects including dilatation by smooth muscle relaxation, inhibition of smooth muscle proliferation, anti-inflammatory effects, and anti-fibrotic action [30]. HF is characterized by insufficient activation of SGC and consequent reduction of cGMP availability resulting in vasoconstriction. Vericiguat directly stimulates SGC and enhances cGMP synthesis independently from NO availability. In contrast to the gliflozins, the discovery of vericiguat followed the classical steps of drug development, setting a priori SGC as the target to be modulated. Actually, vericiguat belongs to a series of soluble SGC stimulators, riociguat being the first-in-class introduced as a novel treatment for pulmonary hypertension [30]. Riociguat, however, has a short half-life, thus it needs to be prescribed three times a day. Studies on structure–activity relationship allowed decreasing oxidative metabolism of riociguat, generating the once-daily modulator vericiguat.

The phase III VICTORIA trial, following the mixed results of the phase II SOCRATES-reduced trial, was a gamble worth taking. The VICTORIA results show that vericiguat lowered the incidence of the composite of CV death and HHF in symptomatic HFrEF (*i.e., LVEF ≤ 45%*) patients [31]. The primary outcome reduction was driven by HHF, as the difference in CV death was non-statistically significant. Tolerability of vericiguat was good, except for, as expected, symptomatic hypotension (*9.1% vs. 7.9%*), syncope (*4% vs. 3.5%*), and anemia (*7.6% vs. 5.7%*), which were more frequent in the treated than in placebo arm [31]. The enrolled patients were well treated. More than a third received a biventricular pacemaker or implantable cardioverter defibrillator (ICD) and 90% two or three anti-hormonal drugs. Compared to PARADIGM-HF and DAPA-HF, the annual rate of the primary outcome in the VICTORIA trial was more than two times higher, reflecting the severity of the enrolled population. Patients with NT-pro BNP in the higher quartile and with LVEF ≤ 35% were those who benefited the most, a setting still to be explored for ARNI and gliflozins.

## 5. Myosin Activators: Omecamtiv Mecarbil

Omecamtiv mecarbil is the first-in-class of selective cardiac myosin activators. It was developed from the knowledge that, physiologically, the binding of myosin to actin filaments is weak until the generation of energy from hydrolysis of ATP to ADP occurs [30]. This is followed by a release of inorganic phosphate (*Pi*) from myosin. The resulting myosin–ADP complex binds tightly to actin and strongly pulls myosin against actin, generating a strong interaction. The complex myosin–ADP–actin dissociates when a new ATP molecule binds again to the myosin head and a new cycle starts. This happens 60 times in a minute for all our life! Of note, during a systole, only 10–30% of the myosin heads interact with actin filaments, leaving ample room for improvement [32]. Omecamtiv mecarbil selectively binds to serine *148*. The subsequent conformational changes increase the speed of the ATP hydrolysis, thus promoting Pi release and the transition from a weak to a strong bond. In so doing, more myosin heads are bound to actin filaments and remain attached for a longer period, thus boosting force production. From a functional point of view, this translates to prolongation of systole duration and increased cardiac contractility. Differently from the other positive inotropes, the mechanism of action of omecamtiv mecarbil is independent from the levels of intracellular calcium and peripheral resistance, without a chronotropic rebound and, importantly, without an increase in myocardial oxygen consumption [33]. In the first small RCT, the Atomic-AHF omecamtiv mecarbil did not improve dyspnea, while the COSMIC-HF trial showed improvement of cardiac function, reduction of ventricular diameters, and of plasma NT-PRO BNP compared to placebo [34,35].

In the GALACTIC-HF trial, omecamtiv mecarbil caused a modest but significant lowering of the combined endpoint of CV death and HF events in patients with symptomatic HFrEF (*i.e., LVEF ≤ 35%*), both in hospitalized or recently dismissed HF patients [36]. The benefit was consistent across all the subgroups with a possible omecamtiv mecarbil was given on top of guideline-recommended HF therapy, with 20% of the population on ARNI [36]. Omecamtiv mecarbil was well tolerated, even in patients with ischemic heart disease and/or ventricular arrythmias. Of note, omecamtiv mecarbil did not induce hypotension even in patients with systolic blood pressure between 85 and 100 mmHg at baseline. Thus, the population of the GALACTIC-HF was similar to that of the VICTORIA trial, but with less drug-induced hypotension.

There were other similarities between the GALACTIC-HF and the VICTORIA trials: (1) in both trials, the efficacy was proved in just a single randomized study; (2) the risk reduction was modest: 10% in the VICTORIA trial and 8% in the GALACTIC-HF trial; (3) the main drivers of the primary outcome were HHF and urgent unplanned outpatient visits for worsening HF. No difference was observed in CV mortality. Nevertheless, the high exposure to HF hospitalizations resulted in an absolute risk reduction similar to those observed in the PARADIGM-HF and DAPA-HF, and a prolongation of the trials likely would have resulted in a reduction in CV mortality [37,38].

## 6. Consequences of the “New Era”

The scenario produced by these RCTs advocates a reconsideration of the therapeutical approach of HFrEF [2,3,4,5,6,7]. Given the different mechanism of action of the new class of drugs and recent detailed subgroup-analysis, there is enough evidence that ARNI and SGLT2 inhibitors are not one the alternative of the other, and should be used at the maximum tolerated dose early in combination with the other anti-hormonal drugs [3,4,5,6,7]. This consideration alone represents a 360 degree change of HFrEF medical treatment. The present care assumes that treatment with ACEi and BB should first be titrated to target dose. This approach is very difficult to follow, especially in elderly patients, and has been criticized as target doses may just be a little more effective than the starting ones [8]. It has been suggested that target effect (*i.e., HR for beta-blockers or ANP levels for ACEi*) instead of target doses should be used. Physicians are authorized to add further anti-hormonal drugs, such as MRA, only when the clinical condition does not improve. After these steps, ARNi, ivabradine, or devices should be considered, according to patients’ characteristics [3,4,5,6,7]. This approach was logical when there was only one “*enemy*” to combat: the excess of neuroendocrine response. Today, the situation is different. Other known or unknown effects, independent from of the neuroendocrine activation, do matter and it would be no longer ethical to deprive HFrEF patients from the benefit produced by the new treatments [13]. Furthermore, if physicians continue to prioritize the achievement of target doses of each drug before starting a new treatment, it will take several months to prescribe all the recommended therapy. This delay is no longer acceptable, considering that both ARNi and SGLT2i start to improve outcome within 30 days from the first administration. Thus, there has emerged the concept of the “*fantastic four*” (*ARNI, beta-blockers, MRA, and SGLT2i*) to be used, unless contraindication, in combination from the beginning of the treatment and at the maximum tolerated dose, without uptitration [5]. In a 55-year-old HFrEF patient, this four-drug approach could prolong life for a further 6.3 years and provide 8.3 years free from CV death and hospital admission [5].

Besides the fantastic quartet, it is important to consider also the other two first-in-class treatments: vericiguat and omecamtiv mecarbil. Thus far, these drugs have not received the same attention of ARNI and SGLT2i as the evidence of their effects comes from one RCT and not from two or more. In addition, the benefit in outcome seems to be less robust than that offered by sacubitril/valsartan and SGLT2i as total or CV mortality was not significantly reduced. However, the studies on vericiguat and omecamtiv mecarbil recruited patients with more advanced HF than those enrolled in the ARNI and SGLT2i trials, and if the studies had been prolonged, a significant reduction in mortality probably would have been achieved [38,39]. These are the reasons why we have considered, at least theoretically, all the four new classes of drugs, knowing that for vericiguat and omecamtiv mecarbil, there are clear limitations and further research is needed before being fully recommended.

## 7. Our Proposal

Our proposal is reported, in a schematic form, in Figure 1, Figure 2 and Figure 3. The figures were built pulling together mechanisms of action, clinical evidence, and inclusion and exclusion criteria used in the RCTs. In this way, it was possible to identify specific treatment options for the most common phenotypes in the real world, such as patients with HF and low blood pressure, hyperkalemia, concomitant kidney disease, atrial fibrillation, and/or the delicate post-discharge period. A degree of flexibility is needed, considering the differences between RCTs and the real world, particularly when related to cut off, which can be respected in RCTs but less so in the daily clinical arena when dealing with older, frail, multimorbid, female patients who are usually excluded from RCTs. Each drug should be used as early as possible and at the maximum tolerated dose.

**Table 1 refers to the baseline characteristics of the patients enrolled in the RCTs and Table 2 to exclusion criteria.** The tables show that the patients enrolled and the criteria used in the various studies were not homogeneous. The most frequent differences related to %EF, renal function, serum K^+^, NT-proBNP, and blood pressure. Data on associations between the new drugs were not available and caution should be taken at this regard.

Figure 1 refers to HFrEF patients with either: 

(**a**) **HFrEF and reduced blood pressure (systolic BP < 100 mmHg)**: there is not a unanimous definition of the meaning of low blood pressure in HF. A systolic blood pressure <90 mmHg is often taken as a limit. Even this assumption might vary, i.e., in patients with proven or underlying CAD, a systolic blood pressure >120 mmHg is suggested. For these reasons, we arbitrarily considered a systolic blood pressure <100 mmHg. In this case, ARNI and vericiguat were not suggested, as a BP <100 was an exclusion criterion in the respective trials. In PARAGON-HF, the effects of sacubitril/valsartan on blood pressure reduction were superior than that of enalapril. The occurrence of symptomatic hypotension (*<90 mmHg systolic BP*) was present in 2.7% of the randomized population vs. 1.4% in the enalapril arm. Equally, hypotension was a side effect of vericiguat. BB, ACEi/ARBs, and gliflozins could also reduce blood pressure but to less extent than ARNI and vericiguat and were normally utilized in asymptomatic patients with BP as low as 95–100 mmHg in view of the offered relevant improvement of outcome. MRA was known to reduce blood pressure in hypertensive patients, but not in those with HF. In RALES, low blood pressure was not an exclusion criterion and in EMPHASIS, patients with blood pressure <85 mmHg were excluded. Therefore, MRA could be safely used. The only two other drugs that had no effect on blood pressure were omecamtiv mecarbil and ivabradine. In both GALACTIC-HF and SHIFT, only patients with a systolic blood pressure <85 mmHg were excluded. 

(**b**) **HFrEF and hyperkalemia** (serum potassium > 5–5.3 mmol/L): under this condition, both ARNI and ACEi were not suggested. In the PARAGON-HF trial, 16% and 17.4% of patients receiving sacubitril/valsartan and enalapril (*respectively*) showed a serum potassium >5.5 mmol/L. It should also be underlined that in PARADIGM-HF, patients with a value of serum potassium >5.2 mmol at screening and of > 5.4 mmol at randomization were excluded. The same was true for MRA, as in EMPHASIS and in RALES serum potassium >5 mmol/L was an exclusion criterion [40,41]. All the other drugs could be used according to the circumstances. 

(**c**) **HFrEF and chronic kidney disease (CKD)**: CKD with glomerular (eGFR < 60 mL/min/1.73 m^2^) is common in HF patients and is a negative prognostic indicator as it carries a double risk for all-cause mortality. As a result, in absolute terms, patients with CKD might benefit the most from treatment with neuroendocrine blockers and/or from the new drugs. It should be considered that eGFR declines with age and more so in patients with HF and may change during the course of HF. Despite this, most RCTs have excluded patients with severe CKD (eGFR < 30 mL/mm/1.73 m^2^), whilst registries show that patients with HF and CKD are undertreated, most likely for fear of worsening CKD. This is wrong because, for example, a sub analysis of the RALES trial showed that spironolactone caused a 30% relative risk reduction for mortality, independently from baseline eGFR, with most benefit in patients with the lowest eGFR. Nevertheless, RALES patients with a creatinine level >2.5 mg/dL were excluded, whilst in EMPHASIS those with an eGFR < 30 mL/min/i.73 m^2^ were not admitted. These are most likely the limits for use of MRA provided that there is no sudden increase in creatinine and serum potassium. The same limitation applied to ivabradine in SHIFT. It is common practice to use ACEi with caution in patients with reduced eGFR and in SOLVD, patients with serum creatinine higher than 2 mg/dL were not randomized. However, it is advised not to stop ACEi/ARRS, but just to reduce the dose in the case that creatinine increases to >3.5 mg/dL or eGFR is reduced to <20 mL/min/1.73 m^2^. Theoretically, caution should be necessary also for gliflozins and ARNI. In the DAPA-HF trial, patients with eGFR < 30 mL/min/1.73 m^2^ and 20 mL/min/1.73 m^2^ in EMPEROR-Reduced were excluded. However, both in CREDENCE and DAPA CKD and also in EMPEROR-Reduced, gliflozins reduced the risk of progression of both CKD and HF, consistent with the benefit on kidney function observed in trials of SGLT2i in diabetic patients who largely did not have HF [28,29]. In the PARADIGM-HF, an eGFR < 30 mL/min/1.73 m^2^ was also an exclusion criterion for both sacubitril/valsartan and enalapril but, in the long treatment, sacubitril/valsartan actually improved kidney function [10,11]. Thus, despite the exclusion criteria, in the first trial, both ARNI and gliflozins were potentially beneficial in patients with CKD. Omecamtiv mecarbil did not have any contraindication. The GALACTIC trial enrolled patients with VGF as low as below 20 mg mL/min/1.73 m^2^. The same applied to vericiguat as, in the VICTORIA trial, it was given to patients with eGFR < 30 mL/min/1.73 m^2^. BB can also be safely used down to eGFR < 30 mL/min/1.73 m^2^. 

(**d**) **HFrEF with increased arrhythmic burden** (*non-sustained ventricular tachycardia -NSVT or repetitive extra systoles at Holter monitoring*): this is not a contraindication for the majority of drugs prescribed in HF. It is an election criterion for BB. CIBIS III showed that early treatment with bisoprolol reduced runs of TVNS and arrhythmic death better than ACEi [42]. Sudden cardiac death was also reduced by sacubitril/valsartan in PARAGON-HF compared to ACEi and by MRA in RALES, and EMPHASIS-HF. Therefore, BBs, MRA, and ARNI are the pharmacological class with documented reduction of sudden death. The same evidence is not available for ivabradine. In SHIFT, patients with ICD who experienced shock in the last six months and those with II° or more degrees of *BRV* were not included. Caution should be applied when using omecamtiv mecarbil which, theoretically, could increase arrythmias as a consequence of the positive inotropism. It is, however, fair to report that in the GALACTIC trials, ventricular arrhythmias were not increased in the treated arms. For gliflozins or vericiguat, there were no data.

Figure 2 refers to patients with high heart rate (*HR*) and/or atrial fibrillation (*AF*). 

(**a**) **HFrEF in sinus rhythm with elevated HR (*> 70 beats/min*).** These patients should receive BBs at target doses. If HR persists >70 beat/min, as often is the case, then ivabradine should be associated to BBs. This combination provided better HR control and better uptitration of BBs. ACEi, ARNI, and vericiguat can be used with some caution as their vasodilatory effect could result in reflex tachycardia. There were no contraindications for gliflozins, MRA, and omecamtiv mecarbil.

(**b**,**c**) **HFrEF with atrial fibrillation (AF) and HR > and < 100 beats/min.** There was no clear indication on the optimal HR in patients with HF and AF. It is common practice to try to maintain HR between 60 and 70 beats/min [6]. In these conditions, the only drug which was not suggested was ivabradine as AF was an exclusion criterion in SHIFT. In addition, RCTs in patients without HF showed that ivabradine could favor occurrence of AF [43,44]. A recent metanalysis has shown that gliflozins’’ use is associated with significant low risk of AF incidence. Despite the fact that these data are very promising, more research is needed to find out whether these drugs can reduce serious adverse events in HF patients with AF [45]. Although counterintuitive, the use of BBs in AF needs clarification. While these drugs are important to control HR in HF patients with AF and, therefore, to improve symptoms, there is no clear evidence for prognostic benefits [46]. In contrast, with patients in sinus rhythm, HR was not a prognostic indicator in patients with HF and AF. Digoxin may be used instead of BBs (*especially if blood pressure is low*) for rate control [6]. The European guidelines for diagnosis and treatment for acute and chronic HF [37] have further expanded the need of optimization of HF therapy in presence of AF. The CASTLE-AF study [47] showed a benefit in terms of ablation of AF in HF patients. Two other studies support the concept in subpopulations of HF patients [48,49]. In view of these results, the guidelines extensively comment on the AF issue in HF and the conclusion is to first consider anticoagulation, then try to eliminate the triggers and optimize HF therapy. In case of persistence of symptoms, ablation therapy is highly recommended. This latter approach is also important from the physiopathologic point of view, as rhythm control has been shown to positively affect LA/LV remodeling.

Figure 3 refers to the post-acute, pre-discharge phase. To be pragmatic, it was important to distinguish whether patients were wet (*with hypervolemia*) or dry.

(1)**Post-acute HF wet:** often, during hospitalization, patients may be stabilized but remain congestive. In this condition, BBs were not suggested despite some positive data in the severely compromised patients enrolled in COPERNICUS [50]. These, however, were dry patients, without hypervolemia. ACEi/ARBS and MRA can be used if blood pressure is >100 mmHg. ARNI have been proven to reduce re-hospitalization and NT pro-BNP levels after discharge. For gliflozins, there were limited data, mainly from sotogliflozin in the SOLOIST trial. Ivabradine had no data for post-acute HF, independently from being dry or wet.(2)**Post-acute HF dry:** gliflozins, and ivabradine also had no data to support or exclude their use. The other “classical” drugs, such as BB, ACEi, MRA, and ARNI can be used, depending on the circumstances. Vericiguat and omecamtiv mecarbil also had no data in the post-acute phase independently from being hypervolemic or not. However, the two respective trials enrolled more severely hospitalized patients than in those for the other classes of drugs and their respective effects, so vasodilatation and increased inotropism might be important.

## 8. Limitations

Our study had several limitations, which, basically, were due to scarce availability of data. This article is not at all intended to be a “*sort of guideline*” or a “*consensus*” on how to treat HFrEF. These have been recently made available [37]. On the contrary, it is meant to update internists on the “*New Era*” for treatment of HF, to show the limited evidence available for the most common phenotype of patients, and to stimulate physicians to make their own decision on the best therapy for each patient without fixed algorithms as guidelines tend to do. Equally, no attempt is made to suggest the ideal drug combinations which, in absence of clear evidence, should be considered on the basis of the mechanism of action of each treatment. We have considered the most utilized drugs in HF. We have not commented on diuretics, digoxin, anticoagulant, and ferric carboxymatose. In addition, we have limited our discussion to patients with chronic and just post-acute HF but not on advanced HF. This later HF population has totally different clinical profiles and therapeutic goals needing more intensive treatment than just drug therapy. Equally, we have not commented on the emerging role of imaging and especially MRI in HF patients.

## 9. Conclusions

The scientific community is fortunate to have four new classes of drugs to treat HFrEF, each acting with an individual mechanism of action. The European guidelines have recently provided information on how to use them [37]. Accordingly, we have provided some suggestions on the preferred drugs for the most common HF phenotypes, based on the mechanism of action and on the criteria used in the RCTs. These thoughts may be useful, on the one hand, to stimulate clinical reasoning according to the characteristics of the patient and, on the other, to consider the intimal biological effects, and the available, although limited, evidence for the four new opportunities.

## Figures and Tables

**Figure 1 jcm-10-04325-f001:**
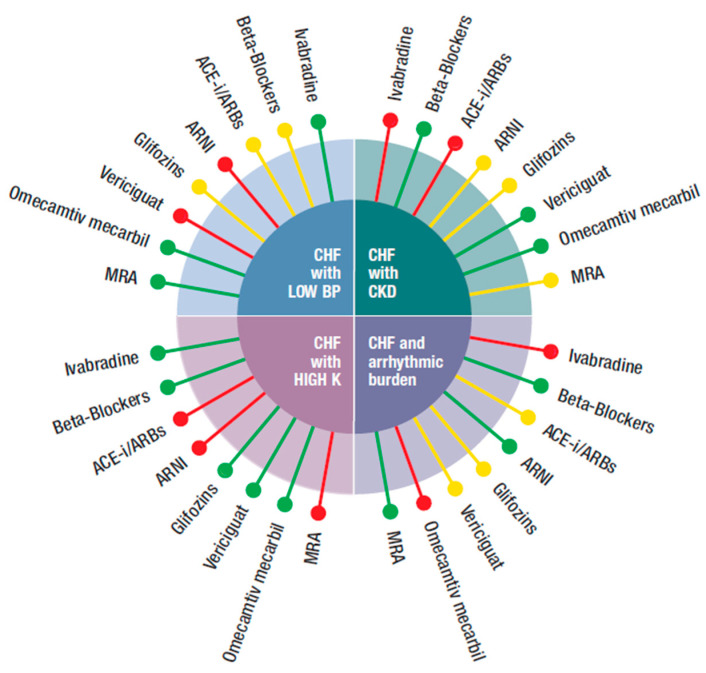
Possible integration of the four new classes of drugs (ARNI, gliflozins, vericiguat, and omecamtiv mecarbil) with the other medications (beta-blockers, ivabradine, ACEi, angiotensin-converting enzyme inhibitor, ARBs, angiotensin receptor blockers, MRA, mineral corticoid receptor antagonist), according to different phenotypes (CHF with: low blood pressure, hyperkalemia, CKD, increased arrhythmic burden) of CHF patients. 
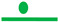
 Preferred, 
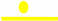
 Possible, 
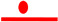
 Not suggested or great caution needed.

**Figure 2 jcm-10-04325-f002:**
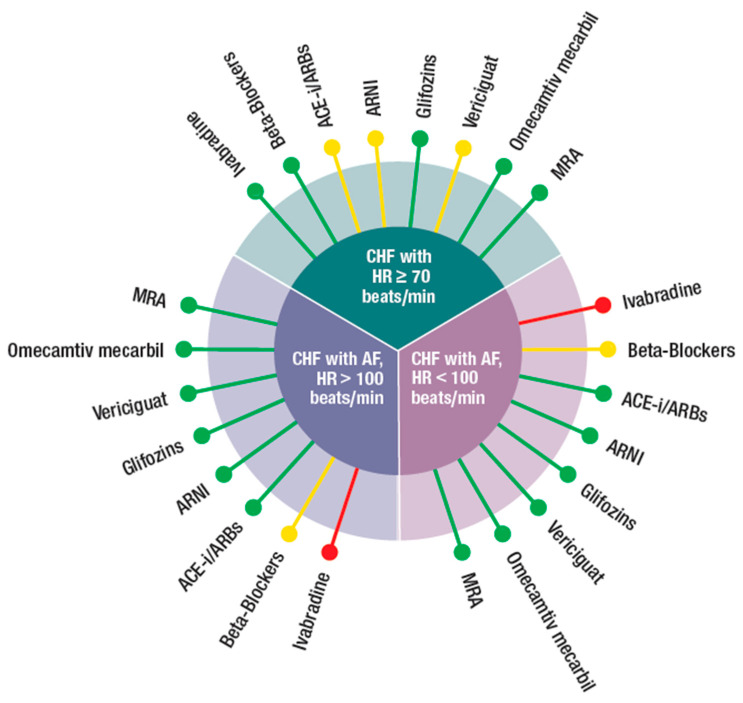
Possible integration of the four new classes of drugs (ARNI, gliflozins, vericiguat, and omecamtiv mecarbil) with the other medications (beta-blockers, ivabradine, ACEi, angiotensin-converting enzyme inhibitor, ARBs, angiotensin receptor blockers, MRA, mineral corticoid receptor antagonist), according to different phenotypes (CHF in sinus rhythm with: HR ≥ 70 beat/min, CHF with AF and HR > 100 or < 100 beats/min) of CHF patients. 
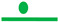
 Preferred, 
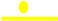
 Possible, 
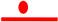
 Not suggested or great caution needed.

**Figure 3 jcm-10-04325-f003:**
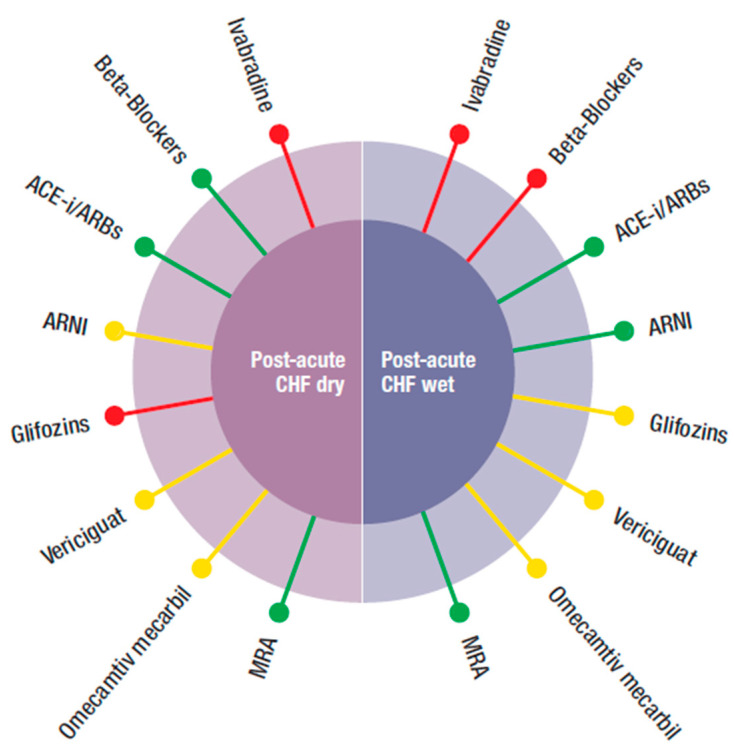
Possible combination of the four new classes of drugs (ARNI, gliflozins, vericiguat, and omecamtiv mecarbil) with the other medications (beta-blockers, ivabradine, ACEi, angiotensin-converting enzyme inhibitor, ARBs, angiotensin receptor blockers, MRA, mineral corticoid receptor antagonist), according to different phenotypes of post-acute HFrEF patients. 
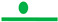
 Preferred, 
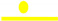
 Possible, 
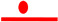
 Not suggested or great caution needed.

**Table 1 jcm-10-04325-t001:** Characteristics of patients at baseline.

	PARADIGM HF (N = 8442)	DAPA-HF (N = 4744)	EMPEROR-reduced (N = 3730)	VICTORIA (N = 5050)	GALACTIC HF (N = 8256)	SOLVD (N = 2569)	CIBIS II (N = 2647)	RALES (N = 1663)	EMPHASIS (N = 2737)	SHIFT (N = 6558)
**AGE (years)**	63.8 ± 11.5	66.2 ± 11	67.2 ± 10.8	67.5 ± 12.2	64.5 ± 11.3	60.7	61 ± 11	65 ± 12	68.7 ± 7.7	60.7 ± 11.2
**MALE (%)**	79	76.2	76.5	76	78.8	80.9	81	73	77.3	76
**BMI (kg/m^2^)**	28.1 ± 5.5	28.2 ± 6	28.0 ± 5.5	27.7 ± 5.8	28.5 ± 6.2	NA	NA	NA	27.5 ± 4.9	28 ± 5.1
**NYHA III-IV (%)**	23.9	32.3	24.9	41.4	46.7	31.8	100	99.6	0	52.0
**NYHA II-III (%)**	94.7	99.2	99.5	98.6	97	88	83	70	100	98
**NYHA IV (%)**	0.8	0.8	0.5	1.4	3	1.9	17	30	0	2
**LVEF (%)**	29 ± 6.1	31.2 ± 6.7	27.7± 6	29 ± 8.3	26.6 ± 6.3	24.8 ± 6.1	27.5 ± 6	25.2 ± 6.8	26.2 ± 4.6	29 ± 5.1
**HR AT ENTRY**	72 ± 12	71.5 ± 11.6	71.0 ± 11.7	73.1 ± 13	72.4 ± 12.2	80 ± 14	79.9 ± 15	81 ± 15	72 ± 12	79.7 ± 9.5
**NT-proBNP (pg/mL)**	1631	1428	1887	2816	1977	NA	NA	NA	NA	NA
**SERUM CREATININE (mg/dL)**	1.13 ± 0.3	NA	NA	NA	NA	1.2 ± 0.3	NA	NA	1.14 ± 0.3	NA
**eGFR (mL/min/1.73 m^2^)**	70 ± 20	66 ± 19.6	61.8 ± 21.7	61.5 ± 27.2	58.8 ± 14.5	65.7 ± 19.0	64 ± 26	65.3 ± 23	71.2 ± 21.9	74.6 ± 22.9
**SERUM POTASSIUM (mmol/L)**	4.5 ± 0.6	NA	NA	4.5 ± 0.5	NA	4.3 ± 0.4	NA	4.29 ± 0.5	4.3 ± 0.4	NA
**AF (%)**	36.2	38.6	35.6	44.9	27.8	11.5	21	27.8	30.8	0
**HISTORY of HHF (%)**	62.3	47.4	31.0 (12 months)	84	100 (12 months)	NA	NA	53.2	52.3	100 (12 months)
**In Hospital Randomization**	0	0	0	NA	25.2	NA	0	NA	0	0

**Table 2 jcm-10-04325-t002:** Inclusion criteria of the trial.

	PARADIGM HF (N = 8442)	DAPA HF (N = 4744)	EMPEROR-reduced (N = 3730)	VICTORIA (N = 5050)	GALACTIC HF (N = 8256)	SOLVD (N = 2569)	CIBIS II (N = 2647)	RALES (N = 1663)	EMPHASIS (N = 2737)	SHIFT (N = 6558)
**AGE** **years**	≥18	≥18	≥18	>18	>18–<85	≤80	18–80	>18	≥55	>18
**NYHA**	II-IV	II-IV	II-IV	II-IV	II-IV	I-IV	III-IV	III-IV	II	II-IV
**LVEF %**	≤40	≤40	≤40	<45	≤35	≤35	≤35	<35	<30	≤35
**creatinine** **mg/dl**	NA	NA	NA	NA	NA	≤2.5	<3.4	<2.5	NA	NA
**Potassium mmol/L**	<5.2	No limit	No limit	No limit	No limit	No limit	No limit	≤5.0	≤5.0	No limit
**SBP** **mmHg**	>100	≥95	≥100	>100	≥85	No limit	≥100	NA	>85	≥85
**eGFR** **mL/min/1.73 m^2^**	>30	≥30	≥20	>15	≥20	NA	NA	NA	≥30	≥30
**NT-proBNP** **(pg/mL)**	≥600; ≥400 (with HHF 12mo)	≥600; ≥900 (IF AF); ≥400 (IF HHF < 12mo);	≥600;* ≥1000;** ≥2500; ***	≥1000; ≥1600 (with AF)	≥400; ≥1200 (with AF)	NA	NA	NA	≥500 M and ≥750 F (without HHF < 6 mo)	NA

* ≥ 600 (EF ≤ 30% or ≤ 40% with HHF in 12 months); The values doubles with AF; ** ≥ 1000 (EF between 31–35%); *** ≥ 2500 (EF > 35%); The values double in case of concomitant AF.

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
