# Peer review of "Looking for a Tailored Therapy for Heart Failure: Are We Capable of Treating the Patient Instead of the Disease?"

_jcm, 2021, doi:10.3390/jcm10194325_

Round 1

Reviewer 1 Report

I agre with  the authors of this statement that after almost a decade of stagnation in clinical research for HF treatment, five large ran- 11
domised trials recently published have supported the use of four new classes of drugs, namely: 12
angiotensin receptor/neprylisin inhibitor, sodium-glucose co-transporters 2 inhibitors, soluble 13
guanylate cyclase modulators, and myosin activators. Each treatment has proved to be beneficial 14
for both long-term outcomes and quality of life. Beside their clinical relevance, all these novel treat- 15
ments have a different mechanism of action beyond the conventional neuro-hormonal blockage. 16
These different pathways, together with the unquestionable clinical evidence, advocate a re-think- 17
ing of HF treatment and of the appropriate drug to integrate with the existing standard therapy, 18
according to different characteristics of HFrEF patients. This review paper aims to provide a syn- 19
thetic overview of the mechanisms of action of the new drugs and to propose a more personalized 20
approach, considering patients’ characteristics and safety profiles. To this end, we have identified 21
seven profiles for patients with chronic heart failure with reduced ejection fraction and two for pre- 22
discharge patients. This is an interesting paper, the Authors may a little be in a conflict with ESC or AHA because this paper, even though they say it is a not a position paper or some kind of a guideline but it looks like a guideline which show how to treat heart failure patients in terms of the use of new drugs. Even though I like this paper, I enjoyed reading it I think it might be highly cited .No other  comments because I think that this manuscript is well written and the design of this papre is very good.

Author Response

We thank the referee for his very kind comments and we are pleased that we are sharing the same view related to the modern treatment of HFrEF in the light of the newly available drugs.

Reviewer 2 Report

This manuscript reviews entitled "Looking for a tailored therapy for heart failure: are we capable 2 of treating the patient instead of the disease?" focuses on the new HF treatments that are or will be soon available to physicians.

The manuscript is well written and represent a rather complete review of current evidence regarding new HF drugs. However I have difficulties with the underlying concept of tailoring therapy. I would rather argue that we probably to need as much tolerated drug as possible. And possibly favor low dose to be able to implement as many drug class as possible. The first glimpse to the guidelines we had at the HFA meeting was heading in this direction with ACEI/ARNi, BB, MRA andSGLT2i being side by side on the first step of management. I consequently am unsure of the angle followed by the authors.

Second, the guidelines will be released in 2 weeks. I am afraid that this manuscript will loose considerable impact at this release. I would rather suggest to reshape the manuscript in light of those new soon to be released guidelines.

Author Response

We thank the reviewer for his comments.

We apologise if the manuscript has provided the impression that multiple treatment should be used in combination at lower doses. This, indeed, was not our intention. To avoid this, we have emphasized the concept of full doses everywhere in the manuscript by adding the highlighted phrases, i.e.:

  • Page 9, lines 459-461 “there is enough evidence that ARNi and SGLT2 inhibitors are not one the alternative of the other, and should be used at the maximum tolerated dose early in combination with the other anti-hormonal drugs”;
  • Page 9, lines 477-480 “Thus, has emerged the concept of the “fantastic four” (ARNi, beta-blockers, MRA, and SGLT2i) to be used, unless contraindication, in combination from the beginning of the treatment and at the maximum tolerated dose, without uptitration”;
  • Page 10, lines 523-524 “Each drug should be used as early as possible and at the maximum tolerated dose without uptitration”.

The general idea was to make available, for general physicians and internal medicine experts who are the main readers of the Journal of Clinical Medicine, which are the preferred (and the contraindicated) drugs according to the most common phenotypes of heart failure. We do believe that this information, mainly based on the mechanisms of action of the new drugs (which may be not fully appreciated by general physicians) may prove to be useful for them even before the release of the guidelines of the ESC.

The second issue raised by raised by the referee relates to the timing of publication in view of the coming release of the European guidelines for heart failure. Again, we agree with the referee and his concern was also our concern. However, we think it is not correct to delay the publication  of our work in view of the approaching release of the guidelines. Our proposal is in no way contradicting the position of the guidelines which, although not yet published, have been already released in Florence, during the meeting of the Heart Failure Association. Actually, our paper, like the guidelines, supports the immediate use of the combination of the four fundamental drugs (ARNI/ARBs, SGLT2i, Beta-Blockers, MRA) and puts an end to the step by step strategy. At the same time, our proposal is trying to find room for a more personalized use of these combinations according to the different mechanisms of action of each drug (which are summarized in a simple way) in the most common phenotypes. We also take into consideration the last two newly discovered drugs, vericiguat and omecamtiv mecarbil as they could be already available in some countries. Also the figures and the table could be useful in case one or two of the four fundamental treatments are not tolerated or cannot be administered at the optimal doses.

For these reasons, an early publication could contribute to the usual post guidelines dialectic with the advantage that it will be highly cited. This is also the opinion of the other referee.